# Benchmarking Deep Inverse Models over time, and the Neural-Adjoint method

**Simiao Ren**

Dept. of Electrical and Computer Engineering
Duke University
Durham, NC 27705
`simiao.ren@duke.edu`

**Willie J. Padilla**

Dept. of Electrical and Computer Engineering
Duke University
Durham, NC 27705
`willie.padilla@duke.edu`

**Jordan Malof**
Dept. of Electrical and Computer Engineering
Duke University
Durham, NC 27705
`jordan.malof@duke.edu`

## Abstract

We consider the task of solving generic inverse problems, where one wishes to determine the hidden parameters of a natural system that will give rise to a particular set of measurements. Recently many new approaches based upon deep learning have arisen, generating promising results. We conceptualize these models as different schemes for efficiently, but randomly, exploring the space of possible inverse solutions. As a result, the accuracy of each approach should be evaluated as a function of time rather than a single estimated solution, as is often done now. Using this metric, we compare several state-of-the-art inverse modeling approaches on four benchmark tasks: two existing tasks, a new 2-dimensional sinusoid task, and a challenging modern task of meta-material design. Finally, inspired by our conception of the inverse problem, we explore a simple solution that uses a deep neural network as a surrogate (i.e., approximation) for the forward model, and then uses backpropagation with respect to the model input to search for good inverse solutions. Variations of this approach - which we term the neural adjoint (NA) - have been explored recently on specific problems, and here we evaluate it comprehensively on our benchmark. We find that the addition of a simple novel loss term - which we term the boundary loss - dramatically improves the NA's performance, and it consequentially achieves the best (or nearly best) performance in all of our benchmark scenarios.

## 1   Introduction

In this work we consider the task of solving generic inverse problems. An inverse problem is characterized by a forward problem that models, for example, a real-world measurement process or an auxiliary prediction task. The forward problem can be written as

$$y = f(x) \qquad (1)$$

where $y$ is the measurable data, $f$ is a (non-)linear forward operator that models the measurement process, and $x$ is an unobserved signal of interest. Given $f$, solving the inverse problem is then a matter of finding an inverse model $x = f^{-1}(y)$. However, if the problem is ill-posed (e.g.,

non-existence, or non-uniqueness, of solutions), finding $f^{-1}$ is a non-trivial task. Specific inverse problems can be solved using apriori knowledge about $f$, (e.g., sparsity in some basis, such as compressed sensing), however, we consider the task of solving generic inverse problems, where no such solutions are known.

Recently many new approaches based upon deep learning have arisen, generating impressive results. These methods typically require a dataset of sample pairs $\{x_n, y_n\}_{n=1}^{N}$ from $f$, from which a deep neural network model can be trained to approximate the inverse model, $\hat{f}^{-1}$. Some recent examples include models based on normalizing flows (e.g., invertible neural networks [1, 2]), variational auto-encoders [3], tandem architectures [4, 5].

## 1.1 Modern inverse models as stochastic search

Despite the apparent variety of recent approaches, most of these inverse models can be written in the form $\hat{x} = \hat{f}^{-1}(y, z)$, where $z$ is randomly drawn from some probability distribution $Z$ (e.g., Gaussian). Although the interpretation of $z$ varies across these models, they all share the property that the $\hat{x}$ returned by the model will vary depending upon the value of $z$. Furthermore, since it is usually trivial and fast to evaluate the accuracy of a candidate inverse solution using the forward model, $f$ (e.g., a simulator), one can search for more accurate inverse solutions by sampling multiple values of $z$, each yielding a different inverse solution. Each solution can then be validated using $f$, and the best solution among all candidates can be retained. Therefore, each modern inverse model can be viewed as a means of efficiently, but nonetheless stochastically, searching through x-space for good solutions.

From this perspective, the performance of each inverse model depends upon the number of $z$ samples that are considered, denoted $T$. For example, one model may perform best when $T = 1$, while another model performs best as $T$ grows. Our experiments here show that this is indeed the case, and model performance (relative to others) is highly dependent upon $T$. Typically however the performance, $r$, of an inverse models is judged by estimating its expected "re-simulation" error [2] over the data and latent variable distributions, denoted $D$ and $Z$ respectively. Mathematically, we have

$$r = E_{(x,y)\sim D, z\sim Z}[\mathcal{L}(\hat{y}(z), y)] \tag{2}$$

where $\hat{y}(z) = f(\hat{f}^{-1}(y, z))$ is the "re-simulated" value of $y$ produced by passing $\hat{x}$ (an estimate) through the forward model, and $\mathcal{L}$ is the user-chosen loss function (e.g., L2 loss). The metric $r$ effectively measures error under the assumption we always utilize one sample of $z$ (given a target $y$). Here we propose an alternative metric that quantifies the expected *minimum* error if we draw a sequence of $z$ values of length $T$, denoted $Z_T$. Formally, this is given by

$$r_T = E_{(x,y)\sim D, Z_T\sim \Omega}\Big[ \min_{z\in Z_T}[\mathcal{L}(\hat{y}(z), y)]\Big] \tag{3}$$

where $Z_T$ is a sequence of length $T$ drawn from a distribution $\Omega$. This measure characterizes the expected loss of an inverse model *as a function of* the number of samples of $z$ we can consider for each target $y$. In this work we conduct a benchmark study of four tasks with $r_T$, and we find that the performance of modern inverse models depends strongly on $T$, revealing the limitation of existing metrics, and revealing useful insights about the way in which each model stochastically searches $x$-space. In particular, we present analysis suggesting that modern inverse models suffer from one or both of the following limitations in their search process: (i) they don't fully explore $x$-space, missing some solutions; or (ii) they do not precisely localize the optimal solutions, introducing error.

## 1.2 The neural-adjoint method

Inspired by our conception of the inverse problem, we explore a simple solution where the main idea is to train a neural network to approximate $f$ and then, starting from different random locations in $x$-space, use $\partial \hat{f}/\partial x$ to descend towards locally optimal $x$ values. Variations of this approach have recently been employed on a few specific problems [6, 7], however, here we evaluate its competitiveness against other modern approaches on several tasks. We also add a novel simple term to its loss function - which we term the boundary loss - that dramatically improves its performance. We call the resulting method the Neural Adjoint (NA), due to its resemblance to the classical Adjoint method for inverse design [8, 9]. Surprisingly, the relatively simple NA approach almost always yields the lowest error among all models, tasks and $T$-values considered in our benchmarks. Our

analysis suggests that, in contrast to other models, NA fully explores the $x$-space, and also accurately localizes inverse solutions. NA achieves this advantage at the cost of significantly higher computation time, which as we discuss, may disqualify it from some time-sensitive applications.

In summary, the three primary contributions of this work are as follows:

1. *A comprehensive benchmark comparison using $r_T$.* We compare five modern inverse models on four benchmark tasks. The results reveal the performance of modern models under many different conditions, and we find that their accuracy depends strongly on $T$.

2. *A new modern benchmark task, and a general method to replicate it.* We introduce a contemporary and challenging inverse problem for meta-material design. Normally, it would be difficult for others to replicate our studies because requires sophisticated electromagnetic simulations. However, we introduce a strategy for creating simple, fast, and sharable *approximate* simulators for complex problems, permitting easy replication.

3. *The neural-adjoint (NA) method.* The NA nearly always outperforms all other models we consider in our benchmark. Furthermore, our analysis provides insights about the limitations of existing models, and why NA is effective.

We release code for all inverse models, as well as (fast) simulation software for each benchmark problem, so that other researchers can easily repeat our experiments. [1]

## 2   Related Work

**Modern deep inverse models**. Given some samples from a forward model, learning the inverse mapping is difficult even for trivial tasks because of one-to-many mappings, where several input values (e.g., designs) all give rise to the same (or similar) forward model output [10]. This causes problems with many optimizers and loss functions because they assume a unimodal output. For example, using gradient descent with mean-squared error causes the model to produce solutions that are an average of all individual solutions, which is usually not a valid solution. To address this inconsistent gradient information, cyclic consistent loss or Tandem models [4, 11, 12, 13] avoid this dilemma by connecting a forward model to the backward model, thereby effectively backpropagating using only one solution, even if multiple solutions exist. An alternative approach is to model the conditional posterior, $p(x|y)$, directly using variational methods [14, 15]. Variational Auto-Encoders (VAEs) [3] consist of an encoder and decoder, and model the joint distribution of hidden and measurement states, to normal distributions z, and decode inverse solutions from samples. By minimizing the evidence lower bound, it trades between reconstruction accuracy and transformed joint distribution closeness to a normal distribution. Earlier work on Mixture density networks (MDNs) [16] directly model the conditional distribution using a mixture of gaussian distributions. The parameters of the gaussians are predicted by a feedforward neural network. With recent advance in the normalizing flow community [17, 18, 19, 1] first applied a state-of-the-art invertible neural network to the inverse problem. Utilizing various invertible network-based architectures, Kruse [2] benchmarked them on two simple inverse problems. It was found that conditional invertible networks (cINN), and invertible networks (INN) trained by maximum likelihood, had the best performance. Many of these models were recently benchmarked on two inverse problems in [2]: VAE, INN,cINN, and MDN. We reproduce their results here, but we add a tandem model and the NA model. We also compare all models on two additional benchmark tasks introduced here (i.e., four total task).

**Inverse model performance metrics**. Although the architecture varies across different studies, the performance metric used in each is largely identical. Nearly all the studies on inverse regression problems uses either Mean Squared Error or Root Mean Squared Error *with only one evaluation*, [1, 2, 4, 10, 13, 14, 15, 7] despite the stochastic nature of some approaches, which can produce different solutions for the same target. Posterior matching is less of a focus in this paper and the Maximum Mean Discrepancy (MMD) score is appended in the supplement.

**Adjoint-based methods**. The adjoint method is a popular approach in control theory and engineering design that relies upon finding an analytical gradient of the forward model with respect to the control-lable variables, and then using this gradient to identify locally optimal inverse solutions. The NA method here also utilizes gradients of the forward model to identify locally optimal inverse solutions,

however, by using a neural network to approximate the forward model (and its gradients) there is no need to derive an analytic expression. Variants of this strategy have also recently been employed by [7] for meta-material design (our inspiration), and [6] in molecule design. We primarily build upon their work by (i) distilling and describing the essential elements of this approach; (ii) introducing the boundary loss, and conducting comprehensive experiments that show it substantially improves the accuracy and reliability of this approach; and (iii) conducting a comprehensive comparison of the resulting approach (the NA method) against other modern models.

## 3   The Neural-Adjoint Method

The NA method can be divided into two steps: (i) Training a neural network approximation of $f$, and (ii) inference of $\hat{x}$. Step (i) is conventional and involves training a generic neural network on a dataset of input/output pairs from the simulator, denoted $D$, resulting in $\hat{f}$, an approximation of the forward model. This is illustrated in the left inset of Fig 1. In step (ii), our goal is to use $\partial \hat{f}/\partial x$ to help us gradually adjust $x$ so that we achieve a desired output of the forward model, $y$. This is similar to many classical inverse modeling approaches, such as the popular Adjoint method [8, 9]. For many practical inverse problems, however, obtaining $\partial \hat{f}/\partial x$ requires significant expertise and/or effort, making these approaches challenging. Crucially, $\hat{f}$ from step (i) provides us with a closed-form differentiable expression for the simulator, from which it is trivial to compute $\partial \hat{f}/\partial x$, and furthermore, we can use modern deep learning software packages to efficiently estimate gradients, given a loss function $\mathcal{L}$.

More formally, let $y$ be our target output, and let $\hat{x}^i$ be our current estimate of the solution, where $i$ indexes each solution we obtain in an iterative gradient-based estimation procedure. Then we compute $\hat{x}^{i+1}$ with

$$\hat{x}^{i+1} = \hat{x}^i - \alpha \left. \frac{\partial \mathcal{L}(\hat{f}(\hat{x}^i), y)}{\partial x} \right|_{x=\hat{x}^i} \tag{4}$$

where $\alpha$ is the learning rate, which can be made adaptive using conventional approaches like Adam [20]. Notice that the parameters of the neural network are *fixed*, and we are only adjusting the input to the network, treating them like model parameters. Our initial solution, $\hat{x}^0$ is drawn from some distribution $\Gamma$. Given some desired $y$, NA iteratively adjusts its estimated solution (beginning with $\hat{x}_0$) until convergence (e.g., $\mathcal{L}$ no longer reduces). This entire process acts as the inverse model for the process, $\hat{f}^{-1}(y, z)$, where $z = \hat{x}^0 \sim \Gamma$. This is illustrated in the right inset of Fig 1. Similar to other approaches, we can draw a sequence of $z$ values and obtain an estimated solution for each one.

And as we show in our experiments in Section 6.1, the NA method yields highly accurate solutions compared to other models, however, at the cost of relatively high computation time. One challenge with this approach is that many initializations either (i) do not finish converging, or (ii) converge to a poor minima. To mitigate this problem, we always extract a thousand solutions, and use the NA's built-in forward model to internally rank-order the solutions and return only the top "T" solutions to be evaluated by the true simulator. As we discuss in Section 6.1, this process only marginally increases the inference time of NA (and all inverse methods we consider) because of efficient parallel processing on GPUs. However, because the NA uses an iterative gradient descent procedure, it is still computationally expensive compared to other methods.

### 3.1   Obtaining good results: the boundary loss

Another challenge with NA is that (unless restricted) it frequently converges to solutions that are outside of the training data sampling domain. As we show in the supplement, this seems to occur because $\hat{f}$ becomes highly inaccurate outside of the training data domain, and (erroneously) predicts that $x$-values in this space will produce the desired $y$ (or a close approximation). As a consequence, these inverse solutions are generally inaccurate, resulting in high error when evaluated with the true simulator. To discourage this behavior we add a simple "boundary loss" term that encourages NA to identify solutions that are within the training data domain, where $\hat{f}$ is accurate. This loss term, denoted $\mathcal{L}_{bnd}$, is given by

$$\mathcal{L}_{bnd} = ReLU(|\hat{x} - \mu_x| - \frac{1}{2}R_x) \tag{5}$$

where $\mu_x$ is the mean of the training data, $R_x$ is its range (for unbounded distributions of $x$, we define the range to be the interval of 95% probability), and ReLU is the conventional neural network activation function. This loss is only added during the inference of inverse solutions. As we show in the supplement, without $\mathcal{L}_{bnd}$ added, the performance of NA decreases substantially. In the supplement we also visualize the NA method, and without, $\mathcal{L}_{bnd}$ on a simple 1-dimensional task, illustrating its effects.

**Some limitations**. Although effective, the form of the boundary loss in eq. 5 assumes that the training data domain is well approximated by a hyper-cube of the form $|\hat{x} - \mu_x| - 0.5R_x$. If this is not true (e.g., the domain is non-convex) then $\mathcal{L}_{bnd}$ may become less effective. Furthermore, the form of $\mathcal{L}_{bnd}$ implicitly assumes $\hat{f}$ is uniformly accurate within the training domain, and drops equally in all directions outside of it. However, the accuracy of $\hat{f}$ will not generally meet these assumptions, and it is unclear how the loss in $\mathcal{L}_{bnd}$ would vary as a function of the uncertainty of $\hat{f}$.

**Relationship to other methods**. *Trust region optimization (TRO)* [21]. The boundary loss and TRO both identify regions where $\hat{f}$ is accurately approximating $f$, called a "trust region", and use this information to guide the search for solutions. However, the NA uses a static trust region that encompasses the whole training data domain, while TRO estimates local trust regions during the solution search process. *Bayesian Optimization (BO)* [22, 23]. In BO, $\mathcal{L}_{bnd}$ can be interpreted as a prior on the credible interval of the surrogate function ($\hat{f}$ in our case) that is uniformly valued within the training data domain, and then grows outside of it. In BO this prior might cause the acquisition function to sample $f$ at these locations and update $\hat{f}$, however in contrast we use it to discourage the acquisition function (gradient descent in our case) from seeking solutions in these regions. *Adaptive sampling* [24]. In adaptive sampling $\hat{f}$ is progressively updated using samples at locations where it is estimated to be inaccurate. In contrast, $\mathcal{L}_{bnd}$ essentially assumes the model is equally accurate throughout the training data domain and, similar to BO, $\hat{f}$ is not updated with samples from $f$.

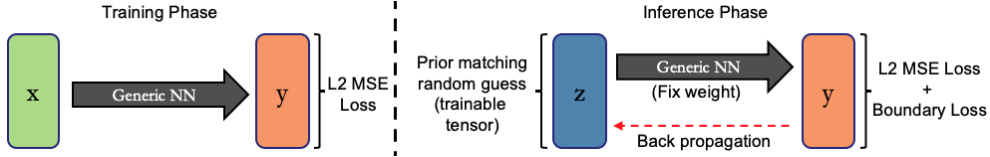

Figure 1: Architecture of Neural Adjoint method

## 4   Benchmark deep inverse models

In this section we briefly describe the inverse models that we employ in our benchmark experiments. We focus on the motivation and relevant properties of each model, however, more detail for each model can be found in the supplement and in referenced supplied for each method.

**Mixture Density Networks (MDN)** [16]. MDNs model the conditional distribution $p(x|y)$ as a mixture of Gaussians, parameterized by $\mu_i, \Sigma_i$ and $p_i$ (mixing proportion). A neural network is trained to predict the *parameters* of the mixture, given a $y$ value, using the following loss:

$$Loss = -\log(\sum_i p_i * |\Sigma_i^{-1}|^{\frac{1}{2}} * \exp(-\frac{1}{2}(\mu_i - x)^T \Sigma_i^{-1}(\mu_i - x))) \tag{6}$$

The number of Gaussians is a hyper-parameter. Once the parameters are predicted for a given $y$, then $\hat{x}$ are inferred by randomly sampling the mixture distribution, and therefore each sample represents a different $z$ value in the stochastic search process.

**Conditional Variational Auto-Encoder (cVAE)**. [14, 15] Created by Kingma [3] it encodes $x$, conditioned on $y$, into a Gaussian distributed random variable $z$. It is a bayesian approach with a proxy loss of Evidence Lower Bound.

$$Loss = (x - \hat{x})^2 - \frac{\alpha}{2} \cdot (1 + log\sigma_z + \mu_z^2 - \sigma_z) \tag{7}$$

$Z$ (re-parameterized into $\sigma_z, \mu_z$) represents the transformed distribution of hidden state $x$ given $y$. The transformation is learned with trade-off between the reconstruction (decoding back to exactly

the same $x$) and distribution ($z$ being normal). cVAE explores the solution space by drawing new examples from $\sigma_z, \mu_z$. We used the implementation introduced by [14] for this approach.

**Invertible Neural Networks (INN)** [1]. Invertible Neural Network are based upon the RealNVP [25], and circumvent the one-to-many mapping problem by padding the (assumed) lower-dimensional $y$-space with some random vector $z$, and then learning a bijective transformation between the $x$ and $y \otimes z$ (i.e., cross-product) spaces. There are two ways of training reported in [2]: (i) a supervised L2 reconstruction loss and a Maximum Mean Discrepancy (MMD) [26]; and (ii) a maximum likelihood estimate (MLE) loss to enforce $z$ to be normally distributed [25]. Since the MLE gives a better solution in the literature [1], we adopt it here, given by

$$Loss = \frac{1}{2} \cdot (\frac{1}{\sigma^2} \cdot (\hat{y} - y_{gt})^2 + z^2) - log|det J_{x \mapsto [y,z]}| \tag{8}$$

where $J$ means the Jacobian of mapping from $x$ to $y \otimes z$ space and $z$ represents the transformed values of $x$. Exploration of the inverse solution space is accomplished by sampling $z$ values from a zero-mean Gaussian distribution. These $z$ values are concatenated to the target $y$ value and passed through the network to obtain an inverse estimate, $x$. INN requires equal dimensionality of $x$ and $y \otimes z$; in cases where this is violated, we follow [1] and pad wth zeros.

**Conditional Invertible Neural Networks (cINN)**. Conditional INNs use a similar network structure as INNs, with a modification that instead of learning the bijective mapping from $x$ to $y \otimes z$ space, it learns the bijective relationship between $x$ and $z$ space under condition $y$. The network is trained under MLE loss as well, with the caveat that $y$ does not appear in the loss function due to conditioning.

$$Loss = \frac{1}{2} z^2 - log|det J_{x \mapsto z}| \tag{9}$$

Here $z$ represents the full transformed distribution of $x$ conditioned on $y$. Exploring inverse solution space also requires sampling different $z$ values. We adopted the original author's implementation in both invertible networks, [2] in order to avoid inadvertent alteration of the comparison condition.

**Tandem model** [4, 5]. In this approach a neural network is first trained to approximate $f(x)$ using a standard regression loss (e.g., squared error). The parameters of $\hat{f}$ are then fixed, and an inverse model $\hat{f}^{-1}(y)$ is pre-prended to $\hat{f}$, and it is trained in an end-to-end manner using backpropagation with the following loss:

$$Loss = (\hat{f}(\hat{f}^{-1}(y)) - y_{gt})^2 + \mathcal{L}_{bnd} \tag{10}$$

This loss measures the re-simulation error of each inferred inverse solution, and therefore $\hat{f}^{-1}$ only needs to learn to identify one of the (potentially many) valid inverse solutions to minimize the loss. As a consequence and, unlike all other inverse models, the Tandem only returns one solution for any given $y$ (i.e., it does not benefit as $T$ grows). In the appendix we also show that adding the boundary loss, $\mathcal{L}_{bnd}$, during training is highly beneficial for the Tandem model.

## 5 Benchmark Tasks

We consider four benchmark tasks, which are summarized in Table 1. Inspired by the recent benchmark study [2], we include two popular existing tasks: ballistics targeting (D1), and robotic arm control (D3). For these two tasks we use the same experimental designs as [2], including their simulator (i.e., forward model) parameters, simulator sampling procedures, and their training/testing splits. All details can be found in [2] and our supplement. The remaining two benchmarks are new, and we describe them next.

### 5.1 A new meta-material benchmark (D4), and a technique for replicating it

The goal of this task, recently posed in [27], is to design the radii and heights of four cylinders (i.e., $x \in \mathbb{R}^8$) of a meta-material so that it produces a desired electromagnetic (EM) reflection spectrum ($y \in \mathbb{R}^{300}$), illustrated in Fig. 2. The input and output are (relatively) high-dimensional and non-linear, and $f(x)$ can only be evaluated using slow iterative EM simulators, requiring significant time and expertise. These challenges are typical of modern (meta-)material design problems, forming a major obstacle to progress. Substantial recent research has been conducted on similar problems (e.g., [14, 7, 4, 28, 29]), making this both a challenging and high-impact benchmark problem.

Problems like this are not suitable as benchmarks due to the computation time, needed domain expertise, and required use of a simulator. It is also insufficient simply to share data from the simulator, due to the need to draw new samples from $f(x)$ when evaluating inverse models. We overcome this problem by generating a large number of samples from our simulator (approx. 40,000), and then training an ensemble of deep neural networks to approximate the simulator. This yields a highly accurate simulator (mean-squared-error of 6e-5) that is fast, portable, and easy to use by others. All of our experiments utilize data sampled from this proxy simulator rather than the original simulator. We hypothesized that the difficulty of our meta-material problem may be undermined because we use the same class of models (neural networks) for both the proxy-simulator and our inverse models. We mitigate this risk by providing a much larger set of training data to the simulator model, and using an ensemble of large and varying models for the proxy-simulator.

## 5.2 The 2-dimensional sinusoidal benchmark (D2)

This benchmark problem consists of a simple 2-dimensional sinusoidal function, of the following form: $y = \sin(3\pi x_1) + \cos(3\pi x_2)$. We included this problem because it had both of the following properties: (i) despite its simplicity, we found it is challenging for most of the deep inverse model; (ii) its 2-dimensional input space allowed us to visualize the solutions produced by each inverse model, and study the nature of their errors. We utilize these properties to gain deeper insights about the inverse models in Section 6.2.

Table 1: Benchmarking datasets outline

| ID | Dataset | Dim(x) | Dim(y) |
|----|---------|--------|--------|
| D1 | Ballistics | 4 | 1 |
| D2 | Sine wave | 2 | 1 |
| D3 | Robotic arm | 4 | 2 |
| D4 | Meta-material | 8 | 300 |

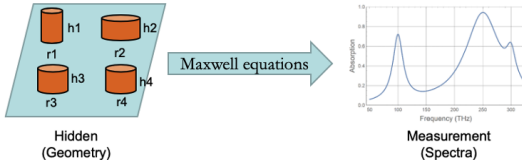

Figure 2: Illustration of the meta-material problem

## 6 Experimental Design and Results

We follow closely the design of the recent benchmark study [2]. For all experimental scenarios that we share with [2], we followed their design and obtained (with one exception) similar results. This includes results for the cINN, INN and cVAE models; on the Robotic Arm and Ballistics tasks. In an effort to compare models fairly, we constrained the newly included models – Tandem and NA – to have the same number (or less) of trainable parameters. Furthermore, all models utilized the same training and testing data, batch size, and stopping criteria (for training). In those cases where model hyperparameters were not available from [2], we budgeted approximately one day of computation time (on common hardware) to optimize hyperparameters, while again constraining model sizes. Full implementation details can be found in the supplementary material.

Once each model was trained, we estimated its error, $r_T$ for $T \in \{1, 10, 20, ..., 50\}$ using $D = \{x_n, y_n\}_{n=1}^N$ random samples from the simulator. We used the following sample estimator of $r_T$:

$$\hat{r}_T = \frac{1}{N} \sum_{n=1}^{N} [\min_{z \in Z_T} \mathcal{L}(\hat{y}(z), y_n)] \tag{11}$$

where $Z_T$ is a randomly drawn sequence of $z$ values of length $T$. We use mean-squared error as $\mathcal{L}$, following convention [2, 1]. A unique set of $z$ values was drawn for each model, based upon the sampling distribution required by that particular model (e.g., Gaussian for cINN).

The main experimental results are presented in Fig. 3. Measuring $\hat{r}_T$ as a function of $T$ yields a much richer characterization of each model's performance compared to using just $T = 1$. In Fig. 3 we see that $\hat{r}_T$ falls steadily as $T$ increases, except for the Tandem model with is not stochastic. Therefore $\hat{r}_T$ quantifies the error one can expect for each model depending upon the computational time/hardware permitted for inference available to a user for their application. Much more interesting is the observation that the performance rank-order of the models also varies with $T$ for all four tasks. Therefore, the best model for a given task (in terms of $\hat{r}$) also depends upon the time/hardware permitted for inference.

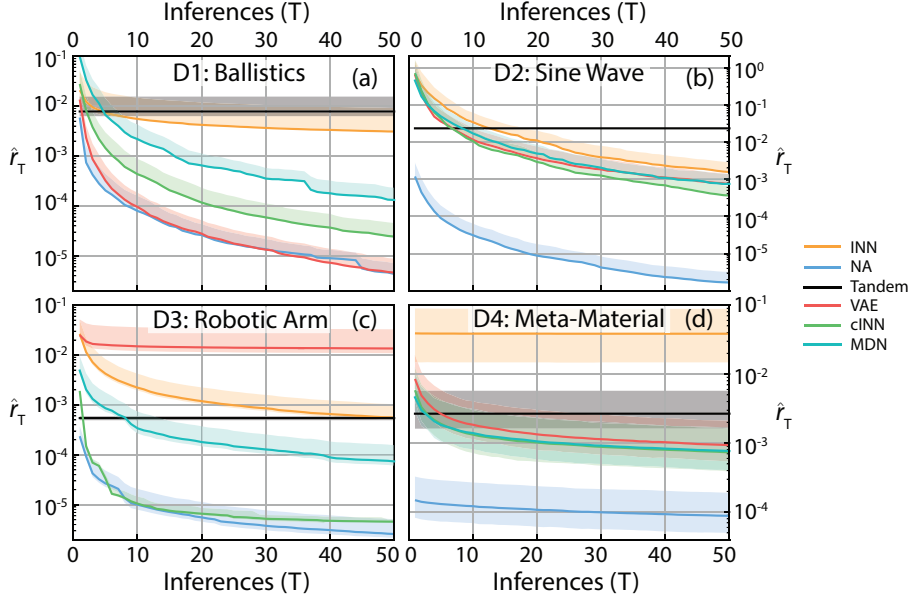

Figure 3: (a-d) Performance on each model for each benchmark task as a function of $T$.

## 6.1 Which models perform best?

The NA method almost always yields the most accurate solutions, across both tasks and settings of $T$. Especially notable is its large performance advantage on the higher-dimensional meta-material task, suggesting it may be especially effective for similar problems. However, NA has the drawback of significantly greater computational costs than the other models, due to its use of gradient descent. The inference time for all model/task combinations is shown in Table 2. We report the time for a single mini-batch of a thousand inferences, which also closely approximates the inference time for a single inference of each model, due to GPU's efficient parallel processing. Therefore, if one inverse solution can be inferred, then (on standard hardware) many inverse solutions can be obtained in roughly the same amount of time, in which case simulation time becomes the biggest bottleneck (i.e., value of $T$).

With these computational considerations in mind, as discussed, if enough time is available for at least one inference of NA, then it is the best choice for nearly every task and setting of $T$. However, for more time-sensitive applications where a single inference from NA is too slow, e.g., many real-time tasks, we are limited to selecting among the other models, which all have (relatively) similar inference time. Given these similarities, the best choice depends more strongly upon the remaining time available for simulation (i.e., value of $T$). In this scenario, the Tandem model consistently achieves the best accuracy for time-sensitive applications, where $T$ is small. If more than a few simulations can be run, then the cINN and the VAE appear to generally achieve the best results: the cVAE performs best on the ballistics task, while the cINN performs best for the robotic arm and sine wave task.

Table 2: Total Inference time (t) in seconds for 1,000 solutions

| Dataset | NA | Tandem | cVAE | INN | cINN | MDN |
|---------|------|--------|------|------|------|------|
| D1:Ballistics | 1.36 | 0.31 | 0.29 | 0.35 | 0.78 | 0.08 |
| D2:Sine wave | 1.22 | 0.19 | 0.19 | 0.19 | 0.20 | 0.53 |
| D3:Robotic arm | 1.12 | 0.19 | 0.31 | 0.21 | 0.23 | 0.62 |
| D4:Meta-material | 46.10 | 0.50 | 0.47 | 0.22 | 0.25 | 0.41 |

Table 3: Estimated Asymptotic Performance of Each Model ($\hat{r}_{T=200}$)

| Dataset | **NA** | Tandem | cVAE | INN | cINN | MDN |
|---------|--------|--------|------|-----|------|-----|
| D1:Ballistics | **2.50e-7** | 7.84e-3 | 2.80e-7 | 2.20e-3 | 1.18e-6 | 6.6e-6 |
| D2:Sine wave | **1.33e-7** | 1.17e-2 | 4.34e-5 | 1.24e-4 | 2.72e-5 | 5.21e-5 |
| D3:Robotic arm | **6.61e-7** | 5.44e-4 | 1.25e-2 | 2.12e-4 | 8.80e-7 | 1.82e-5 |
| D4:Meta-material | **6.67e-5** | 2.53e-3 | 5.49e-4 | 3.83e-2 | 4.45e-4 | 5.15e-4 |

## 6.2 Why does the neural-adjoint perform so well?

Notably in Table 3, we see that NA always achieves the lowest asymptotic error as a function of $T$, while the other models asymptote at varying levels. Why are the other models limited in their accuracy, even as $T \rightarrow \infty$? One potential explanation is that they do not fully explore $x$-space, and thereby miss some accurate solutions. Alternatively, perhaps they can find solutions near all of the global optima, but they cannot accurately localize them (e.g., their estimates are noisy). To answer this question, we visualize the 2-dimensional sinusoid task (D2), on which most of the models perform poorly. Fig. 4 presents a random sample of inverse solutions produced by each model, laid on top of a 2-dimensional error map of $x$-space (darker is better). The blue rings indicate the optimal solutions for a target measurement of $y$=-0.3. We can see clearly that NA finds highly accurate solutions in each of the globally optimal rings. The cVAE and the cINN seem to find solutions near all of the globally optimal solutions, however they rarely infer perfectly accurate solutions. Therefore both the cVAE and cINN seem to suffer from noisy solutions, rather than inability to find the solutions. Finally, the INN

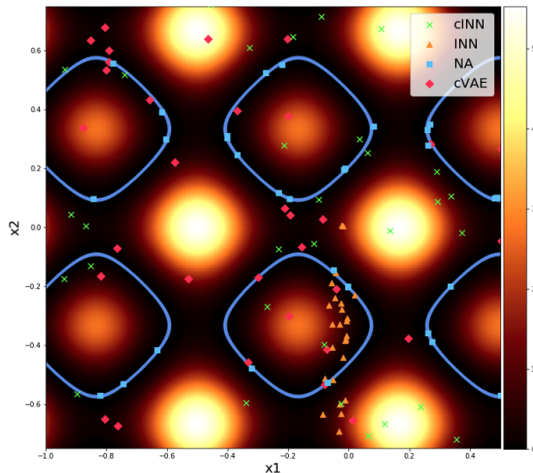

Figure 4: The image axes represent a uniform grid of potential inverse solutions, $(x1, x2)$, for the 2-dimensional sinusoid problem. The pixel intensity at each $(x1, x2)$ location represents the corresponding simulation error of that solution, if our target measurement is $y = -0.3$. The blue rings represent the optimal solutions.

seems unable to search the entire space, in addition to suffering from inaccurate solutions. However, this is a single visualization, representing a single task and a single instance of training the models. We find the relative performance of all models (except NA) varies substantially across tasks *and* the success of their training (which is somewhat random). This suggests that each of these models sometimes suffer from limited exploration of $x$-space, and varying accuracy, depending upon the aforementioned factors. These findings are consistent with (e.g., [2]) overall.

## 7 Conclusions

In this work we presented a large benchmark comparison of five modern deep inverse models, on four benchmark tasks. We propose a new metric, $r_T$, that evaluates the error of models as a function of the number of inverse solutions they are permitted to propose, denoted $T$. We find that the performance of inverse models, both in absolute error and in their rank-order, depends strongly on $T$, suggesting that $r_T$ is important to characterize inverse model performance. We also introduce a challenging contemporary inverse problem for meta-material design. Normally, it would be difficult for others to replicate such real-world problems however, we introduce a strategy for creating simple, fast, and sharable *approximate* simulators. Finally, we propose a method called the Neural-Adjoint, which nearly always achieves the lowest error across all tasks and values of $T$. Its performance advantage is especially strong for the higher-dimensional meta-material problem, suggesting it is a promising approach to solve such problems.

## Broader Impact

We believe the most proximate impacts of this work will be positive. In particular, higher-dimensional inverse problems like our meta-material problem present a major obstacle to the development of beneficial technologies across many disciplines e.g., in materials, chemistry, and bio-chemistry. The Neural-Adjoint method represents a tool to develop much more accurate inverse designs for these complex problems. Furthermore, the ability to replicate inverse studies for complex problems, as we propose, will also accelerate progress, and enable many researchers to study these problems even if they lack sophisticated simulation equipment or expertise. As with many tools, we also acknowledge that these advances can be used to accelerate the development of technologies that are used for negative purposes, which we believe is the most immediate negative outcome of our work.

## Acknowledgments and Disclosure of Funding

We gratefully acknowledges support from the Department of Energy (DOE) (DE-SC0014372).

## Footnotes

[1]`https://github.com/BensonRen/BDIMNNA`

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
