[Supplementary Material]

# Supplementary material: Benchmarking Deep Inverse Models over time, and the Neural-Adjoint method

**Simiao Ren**

Dept. of Electrical and Computer Engineering
Duke University
Durham, NC 27705
simiao.ren@duke.edu

**Willie J. Padilla**

Dept. of Electrical and Computer Engineering
Duke University
Durham, NC 27705
willie.padilla@duke.edu

**Jordan Malof**
Dept. of Electrical and Computer Engineering
Duke University
Durham, NC 27705
jordan.malof@duke.edu

## 1 Posterior matching score

Although the performance over time is the main performance that we want to benchmark, as pointed out by [3] the posterior matching is another metric to measure how good the inverse models are. Below we show the posterior matching score using Maximum Mean Discrepancy (MMD) as a measurement of how close the inferred posterior density is comparing with the ground truth (rejection sampled) distribution. Note that for a real-life problem (D4: meta-material) with higher dimensionality, the rejection sampling becomes intractable. The 3 MMD kernel used was 0.05, 0.2 and 0.9. The code is also available on the repository.

Table 1: Posterior matching MMD score

| Data | NA | TD | cVAE | INN | cINN | MDN |
|---|---|---|---|---|---|---|
| D1:Ballistics | 0.07 | 2.62 | 0.07 | 2.03 | 0.04 | **0.04** |
| D2:Sine wave | 0.04 | 2.84 | 0.03 | 1.07 | 0.03 | **0.03** |
| D3:Robotic arm | 0.06 | 2.70 | 1.62 | 0.11 | 0.04 | **0.03** |

The bold faced models are the best performing ones with respect to posterior matching within one dataset (more digits are compared if ties). We find that Mixture Density Network (MDN) always has the best (lowest) posterior matching MMD score, closely followed by NA and cINN. We hypothesize that MDN wins due to it explicitly optimizes its network weights on posterior matching. Note that the posterior matching does not guarantee good inverse solution on average, as illustrated by the fact that the average re-simulation accuracy of MDN is actually far from that of NA.

## 2 Neural Adjoint (NA) ablation studies

In this section we present an ablation study for (i) the boundary loss, $\mathcal{L}_{bnd}$, and (ii) the design of the distribution from which we draw initial z, $\Omega$ in the Neural-Adjoint method. Our goal is to show

experimental evidence that these additions to NA generally improve its performance. Note that the details of these two methods are provided in the main manuscript. In our ablation experiments we evaluate the performance of NA as we remove or include each of these steps, as shown in Table 2 (left-most three columns). In those cases where we do not design $\Omega$, we set $\Omega$ to a uniform sampling distribution. Aside from the specific experimental variables listed in Table 2, these experiments all follow the experimental design outlined in the main paper.

Table 2: Ablation Study Experimental Design

| Label | Add $\Omega$? | Add $\mathcal{L}_{bnd}$? | $\hat{r}_{200}$ for D1: ballistics | $\hat{r}_{200}$ for D3: robotic arm |
|-------|---------------|---------------------------|------------------------------------|-------------------------------------|
| E1 | No | No | 4.78e0 | 1.53e-6 |
| E2 | No | Yes | 3.54e-6 | 8.87e-7 |
| E3 | Yes | No | 1.39e0 | 5.70e-6 |
| E4 | Yes | Yes | **2.54e-7** | **6.61e-7** |

We conduct these four experiments on two tasks: the ballistics task (D1) and the robotic arm control task (D2) and Fig. 1 presents the results of these experiments in terms of $r_T$ as $T$ varies from one to fifty. The asymptotic performance of each model is estimated by $r_{T=200}$, and is presented in Table 2 (right-most two columns). The results indicate that adding both steps to NA (i.e., transitioning from E1 to E4) results in substantial performance improvements for both tasks considered. For the ballistics task (D1) there is a reduction in error by several orders of magnitude, and for the robotic arm task (D2) there is a reduction in error of 1-1.5 orders of magnitude. The most important step appears to be the boundary loss which, by itself, results in substantial performance improvements. Adding the $\Omega$ design to the boundary loss results in a smaller, but consistent performance improvement. Interestingly we find that adding the $\Omega$ design without $\mathcal{L}_{bnd}$ is detrimental on both tasks.

Figure 1: Effect for with or without Boundary loss and prior initialization

## 3 Neural Adjoint (NA): Visualization of why boundary loss helps

As illustrated in the main paper, the neural adjoint has implementation caveat where constraining the boundary of the solution search phase is crucial to the performance of the inverse problem solving. Here to visualize how and why boundary constraint plays such an important role, we would use a simple toy example of fitting a 1d sine wave (x from $-\pi$ to $\pi$) using a small neural network.

As shown in fig 2a it fits pretty well in the range $[-\pi, \pi]$ and seem to choose an arbitrary small value for out of range domain $(-\infty, -\pi] \cup [\pi, \infty)$. Then we do a neural adjoint method searching for the x value for given y value 0.6 and visualize the error surface of the search in fig 2b. The figure shows how initial guesses of points would be guided by gradient and move towards lower points on this loss surface graph. As several points in the right half found the global minimum, the point initialized to the left struggles to find a lower result. It seems that the left point's failure to find the global minimum

(a) Approximating a simple 1d sine function (x in unit of $\pi$)

(b) Error surface in searching for solution y=0.6

would not cause trouble as we can choose those points who have a lower error in a parallel run to avoid points stuck in local minimum like this.

However, if we run the whole experiment one more time, it is a different story. As shown in fig 3a, the orange line still symbolizes what the neural network learns and the blue line is the ground truth. The blue points are various inverse solution found corresponding to their ground truth point in orange. All those solutions found are deemed global minimum with 1e-1 of error seen by the network. this time the network "decided" not to learn close-to-0 values for out-of-range $(-\infty, -\pi] \cup [\pi, \infty)$ domain. Instead, it learns an affine function at both tails and that unpredictable behavior caused the trouble as some of the solutions are actually out-of-range, thus meaningless and cause a high re-simulation error (0.7).

(a) Prediction chart of 1d sine wave experiment run 2

(b) After adding boundary error during inference in 1d sine wave experiment run 2

As said in the main paper, we solved this with an extra boundary loss which bound the inference solution exploration within the defined range. With the added loss, illustrated in fig 3b, the out-of-range inverse solutions disappear and the re-simulation error converges to the forward approximation error (4e-3).

# 4 Tandem model ablation study

Due to the success of the boundary loss, $\mathcal{L}_{bnd}$, within the Neural Adjoint approach, we also considered whether it may be beneficial for the Tandem model as well. To test this, we evaluate the performance of the Tandem model with, and without, the inclusion of $\mathcal{L}_{bnd}$. We conduct these two experiments on two tasks: the ballistics task (D1) and the robotic arm control task (D2). Aside from excluding the boundary loss, we use the same experimental design used in the main paper. The results of the experiments are presented in Fig 4. the results indicate that inclusion of $\mathcal{L}_{bnd}$ has substantial and consistent benefits, reducing error by at least 2.5 orders of magnitude on both tasks.

Figure 4: Effect for with or without Boundary loss for Tandem model

# 5 Benchmark Deep Inverse Models: Additional Details

In this section we provide additional technical details for each of the existing benchmark inverse models employed in our main paper.

## 5.1 Conditional Variational Auto-Encoder (cVAE)

The conditional variational auto-encoder adopts the evidence lower bound as it encodes the x into gaussian distributed random variable z conditioned on y. During training phase it also make use of the L2 MSE loss to ensure a good reconstruction of the original input x. During inference phase, inverse solution x is decoded from random samples are drawn from z space conditioned on y.

Figure 5: Architecture of conditional Variational Auto-Encoder method

With the evidence lower bound loss defined in equation 1, it trades-off between the reconstruction of the original signal and the shape of the distribution of the latent variable z. Upon minimization of the loss the network is supposed to fully represent the joint distribution using a normally distributed latent space.

$$Loss = (x - \hat{x})^2 - \frac{\alpha}{2} \cdot (1 + log\sigma_z + \mu_z^2 - \sigma_z) \tag{1}$$

## 5.2 Invertible neural network (INN)

The invertible neural network is specially designed to have hard invertibility (full reconstruction). During training, it uses Maximum likelihood loss to map the bigger x space into y and z space, where

z is sampled from a normal distribution. During the inference phase, a randomly drawn normal distributed z would join y to be inverted back to the inverse solution x.

Figure 6: Architecture of Invertible Neural Network method

By taking the assumption that y is normally distributed around its ground truth value, the network can be trained using simple maximum likelihood loss defined in equation 2. To ensure the invertibility the Jacobian of the transformation is also added to the loss, encouraging full invertibility upon convergence. The variance $\sigma$ is set to be small to encourage accuracy and is chosen based on cross validation.

$$Loss = \frac{1}{2} \cdot (\frac{1}{\sigma^2} \cdot (\hat{y} - y_{gt})^2 + z^2) - log|det J_{x \mapsto [y,z]}| \tag{2}$$

## 5.3 Conditional invertible neural network (cINN)

The conditional invertible neural network uses a similar structure as an invertible neural network. Instead of mapping x to yz space, by conditioning on y, it approximates the full mapping between x and a normally distributed random variable z using maximum likelihood as well. During inference, a normally distributed random variable would be drawn to get inverse solution x conditioned on y.

Figure 7: Architecture of conditional INN method

Very much like the Invertible neural network above in equation 2, the conditional version use equation 3 as loss function since it already have y information given.

$$Loss = \frac{1}{2}z^2 - log|det J_{x \mapsto z}| \tag{3}$$

## 5.4 Mixture density network (MDN)

Proposed by [2], Mixture density network provides a simple model for one-to-many relationships by assuming a gaussian mixture for the posterior density where the mean and maximum of the gaussians are determined by the input y. The number of gaussian mixtures is part of the hyperparameters of the network and is tuned by cross validations.

Figure 8: Architecture of MDN method

It is trained using a maximum likelihood method and during inference a guassian sampling is done to retrieve the estimate of x.

$$Loss = -\log(\sum_i p_i * |\Sigma_i^{-1}|^{\frac{1}{2}} * \exp(-\frac{1}{2}(\mu_i - x)^T \Sigma_i^{-1}(\mu_i - x))) \tag{4}$$

# 6 Benchmark tasks: additional details

In this section we provide additional technical details for each of the benchmark tasks included in the main paper, except for the 2-dimensional sinusoid task, due to its simplicity. Two of our tasks are adopted directly from the recent deep inverse model benchmark study [3]: the ballistics task and robotic arm control task. The full details of these benchmarks can be found in [3] but we reproduce them here for completeness.

## 6.1 The ballistics task

A physically motivated dataset as a ball is thrown from position $(x_1, x_2)$ with angle $x_3$ and velocity $x_4$ and land on ground at location $y$. There is no closed form mapping as getting y from given x requires the solve the below equation. Parameter priors are as follows: $x_1 \sim \mathcal{N}(0, \frac{1}{4})$, $x_2 \sim \mathcal{N}(\frac{3}{2}, \frac{1}{4})$, $x_3 \sim \mathcal{U}(9°, 72°)$ and $x_4 \sim \text{Poisson}(15)$.

$$T_1(t) = x_1 - \frac{v_1 m}{k} \cdot (e^{-\frac{kt}{m}} - 1)$$
$$T_2(t) = x_2 - \frac{m}{k^2} \cdot ((gm + v_2 k) \cdot (e^{-\frac{kt}{m}} - 1) + gtk)$$
$$y = T_1(t^*) \; s.t. \; T_2(t^*) = 0$$

## 6.2 The robotic arm control task

Raised by [1], it is a simple geometrical problem asking for the starting height $x_1$ and three joint angles $x_{2,3,4}$ given the robotic arm's final position $[y_1, y_2]$. The closed form relationship is as follows with $l_{1,2} = 0.5, l_3 = 1, \mathbf{x} \sim \mathcal{N}(0, \sigma^2)$ where $\sigma^2 = [\frac{1}{16}, \frac{1}{4}, \frac{1}{4}, \frac{1}{4}]$.

$$y_1 = l_1 sin(x_2) + l_2 sin(x_3 - x_2) + l_3 sin(x_4 - x_3 - x_2) + x_1$$
$$y_2 = l_1 cos(x_2) + l_2 cos(x_3 - x_2) + l_3 cos(x_4 - x_3 - x_2)$$

## 6.3 Meta-material task and approximated simulator

Our meta-material (MM) task is follows the recent work in [4], where the goal was to choose a set of geometric parameters for a MM design so that the resulting MM exhibits some desired electromagnetic properties. In our context, MMs consist of a surface (e.g., a semiconductor wafer) with small repeating geometric structures (e.g., cylinders, crosses) placed on its surface. The characteristics of these structures (e.g., shape, size, thickness) influence the electromagnetic properties of the resulting MM. Our particular MM is composed of a repeating "super-cell" of four cylinders, each with two parameters that we can control: a height and a radius. The electromagnetic property we wish to control is called the reflection spectrum, which is a 300-dimensional vector of values between zero and one. Each value of the reflection spectrum indicates the proportion of signal energy (at each frequency) from an incident electromagnetic ray that would be reflected from the MM surface. In this work our reflection spectrum consists of 300 uniformly-spaced measurements across the frequency range from 0.8 to 1.5 THz, following [4].

While there is currently no known closed-form mathematical expression for the forward model of this system, $f$, we can still evaluate $f$ for a given $x$ using electromagnetic simulation software. Following [4], we use the CST Studio simulation software for this purpose. Although CST is a powerful tool that enables us to study this problem, there are two significant difficulties with using CST (or similar simulators) when studying inverse problems. These two difficulties impede our study, and also

prevent others from replicating our experiments. First, setting up the simulations requires substantial domain expertise that will not be easily accessible to most researchers. The second problem is that evaluating $f$ is relatively slow. Like many simulators, CST evaluates the forward model by iteratively solving a differential equation, in our case Maxwell's equation, which is a relatively slow process. For our particular application, CST can produce (approximately) 1000 simulations per day on a single CPU core, which is about 1 simulation every 1.5 minutes. To carry out our experiment we have 1,000 test points ($y$ values), and we extract 100 proposed solutions ($x$ values) from 4 models for each test point, resulting in 400 days of simulations!

To overcome this problem, following [4], we trained a neural network to closely approximate the CST simulator. Although this approach still required substantial computation time and expertise, overall it required far fewer simulations to generate the data needed to train our "neural simulator" than our inverse modeling benchmarks. Furthermore, we only needed to perform this procedure once, after which we can conduct our experiments much faster using the neural simulator. In addition to being fast, the neural simulator requires little expertise to use by other researchers, making it both fast and easy to use. Therefore the neural simulator approach enables other researchers to easily study this previously inaccessible modern inverse problem. Many important modern inverse problems in engineering and research rely on simulators with the same limitations, preventing widespread study of many problems and slowing scientific progress. We propose this approach as a general strategy to make these complex modern inverse problems accessible to the broader scientific community.

Figure 9: Random Samples from Neural Simulator spectra

For our particular neural simulator, we randomly sampled 40,000 geometry values from a uniform distribution as discussed in [4]. We then used CST to generate corresponding reflection spectra for each of these geometry. Due to inherent symmetry in the parameterization of the meta-material geometry, one can identify several values of $x$ that all correspond to the exact same physical layout of the MM (not discussed [4]). Leveraging this symmetry we were able to expand the total dataset to 160,000 samples without running additional simulations. We split the resulting dataset into two subsets: $80\%$ for training and $20\%$ for testing. Our neural simulator is composed of an ensemble of deep neural network regression models with varying architectures. After training, our neural simulator achieves a mean-squared error of 6e-5 on the test set. Some randomly drawn test spectra are presented in Fig. 9, along with the predicted spectra from the proxy simulator, providing a qualitative illustration its accuracy. The proxy simulator is extremely fast, capable of producing thousands of

forward model evaluations per second. We subsequently used the proxy simulator to generate all of the data in our experiments. We release the proxy simulator with this publication.

# 7 Experimental design: additional details

As discussed in the main body of the paper, our experimental design is based closely upon the recent benchmark study in [3]. In particular, we shared two benchmark tasks (Ballistics and Robotic Arm) and three inverse models (cVAE, INN, and cINN) with the study in [3]. For these particular scenarios we followed their task design, and deep model designs (e.g., architectures and hyperparameters) in all cases in which it was specified. We were able to obtain largely similar results for these common scenarios.

Table 3 presents several additional model training details that we used. These training details remained fixed across all tasks and all models in our experiments. We found that these settings allowed all models to converge before training stopped. With these settings we also were able to find similar error rates to those of [3] on those scenarios that were shared between this work and their work.

For each model and dataset combination, we allocated one day of GPU processing time to optimize the model. For those model/task combinations from Kruse, we did not optimize all model parameters that were already specified. We optimized the remaining parameters (e.g., regularization, multi-task loss weights) but we found these had little impact on our results. For the remaining models/tasks that were not specified in [3] we also considered optimizing model architectures, while remaining within the same overal processing budget. All specifications for our models, and the code used to train them, will be published with our paper.

Table 3: Table for experimental setups

| Parameters | value |
|---|---|
| Training Epoch | 500 |
| Batch size | 1024 |
| Optimizer | Adam |
| Learning rate | 1e-3 |
| Learning rate schedule | half when plateau |
| Optimization time | 1 GPU*day |
| GPU | NVIDIA 1080 Ti |

# 8 Additional miscellaneous results

These additional details were not specifically cited or referenced in the main body of the paper, but we provide them here to supplement the paper.

## 8.1 Average re-simulation error (T=1) Performance

Due to limited space, we did not include the numerical $\hat{r}_T = 1$ in the main paper. The $\hat{r}_T = 1$ is illustrated in table 4.

Table 4: Estimated Average Performance of Each Model $\hat{r}_T = 1$

| Dataset | NA | Tandem | cVAE | INN | cINN | MDN |
|---|---|---|---|---|---|---|
| D1:Ballistics | **5.00e-3** | 7.84e-3 | 1.35e-2 | 2.09e-2 | 2.78e-2 | 9.88e-2 |
| D2:Sine wave | **1.18e-3** | 2.31e-2 | 7.56e-1 | 6.70e-1 | 6.45e-1 | 4.46e-1 |
| D3:Robotic arm | **2.00e-4** | 7.00e-4 | 2.51e-2 | 2.66e-2 | 2.01e-3 | 4.81e-3 |
| D4:Meta-material | **2.50e-4** | 2.53e-3 | 8.60e-3 | 3.89e-2 | 5.70e-3 | 4.60e-3 |

## 8.2 Model size

To cross-validate our result with Kruse [3] , we used models of similar size in the two benchmark problems that we share. For other models and datasets, we decided the size of the model by doing a hyper-parameter swiping and chose best performing model complexity. As shown in Table 5 and Table 6, invertible structures tends to have a larger number of parameters to model complicated invertible relationships while the NA method, as it only needs to model the one-to-one relationship, requires substantially smaller network structures.

Table 5: Model Size in number of free parameters (Millions)

| Dataset | NA | Tandem | cVAE | INN | cINN | MDN |
|---|---|---|---|---|---|---|
| D1:Ballistics | **0.5** | 0.5 | 3.0 | 3.2 | 3.2 | 3.0 |
| D2:Sine wave | **0.7** | 1.5 | 2.5 | 2.1 | 5.3 | 4.0 |
| D3:Robotic arm | 0.8 | **0.3** | 3.0 | 2.6 | 3.2 | 1.5 |
| D4:Meta-material | **3.1** | 3.4 | 19.0 | 7.2 | 11.8 | 6.0 |

Table 6: Saved Model Size (Mb)

| Dataset | NA | Tandem | cVAE | INN | cINN | MDN |
|---|---|---|---|---|---|---|
| D1:Ballistics | **2** | 3 | 14 | 15 | 15 | 14 |
| D2:Sine wave | **3** | 7 | 12 | 9.7 | 24 | 19 |
| D3:Robotic arm | 3 | **2** | 7 | 12 | 15 | 7 |
| D4:Meta-material | **15** | 16 | 88 | 54 | 56 | 31 |

## 8.3 Model training time comparison

The training time for each algorithms are reported under single NVIDIA 1080 GTX GPU. From Table 7 one can see clear trend that NA method method tends to need less training time, which is expected due to their smaller model size.

Table 7: Training Time (s)

| Dataset | NA | Tandem | cVAE | INN | cINN | MDN |
|---|---|---|---|---|---|---|
| D1:Ballistics | **86** | 168 | 155 | 345 | 987 | 187 |
| D2:Sine wave | **82** | 135 | 110 | 191 | 291 | 241 |
| D3:Robotic arm | **70** | 127 | 227 | 932 | 663 | 120 |
| D4:Meta-material | 224 | 256 | 540 | 335 | 882 | **122** |