[Reviews · NeurIPS 2020]

Review 1

Summary and Contributions: The paper considers deep learning for solving inverse problems. It compares existing approaches on several benchmark tasks, including one for metamaterial design. The paper proposes a "neural-adjoint" method which "uses a deep learning model to approximate the forward model, then uses backpropogation to search for good inverse solutions". This method is demonstrated to have best or best-equal performance on most tasks as a function of number of inferences, at the cost of increased time per inference.

Strengths: Comparing of a number of deep inverse models on a number of benchmarks, as a function of the number of proposed solutions, is great idea. The proposed "neural adjoint method" is a smart approach for inverse problems and the experimental evidence convincingly supports the superior performance of this to the other inverse problem models considered.

Weaknesses: Given many of the benchmarks presented here were also presented in "Benchmarking invertible architectures on inverse problems", the value of the benchmarks presented seems somewhat marginal. I have a concern about the paper's presentation of the proposed "neural adjoint method", which seems to me to be a straightforward application of techniques from NN surrogate-based / model-based optimization to inverse modeling. I discuss this concern in "relation to prior work" below. The metamaterial task is very interesting, and seems like a good benchmark: however this was proposed in a "Deep learning for accelerated all-dielectric metasurface design" and so cannot be claimed as a contribution by the current paper. To the authors: does the current paper make a substantial novel contribution, beyond this prior work, to this task as a benchmark?

Correctness: I believe the claims to be correct. I have not checked the code but the experimental design looks thorough and the experimental results look plausible.

Clarity: The paper is well written and easy to read.

Relation to Prior Work: "Inverse problems" can be framed as optimization, minimizing the loss L(x) where this loss is a distance between yhat = f(x) and the observations y. Thus I have a potential case with the paper's presentation of the "neural adjoint" method as related to previous work There is lots of work on using NNs for model-based or surrogate-based optimization. Sometimes people model an objective function Jhat = ftheta(x), and search (i.e. via gradient descent) for x* which minimizes Jhat: this is most common in Bayesian optimization (e.g. see "Scalable Bayesian optimization with neural networks"). Sometimes people model an output yhat = ftheta(x), and search (i.e. via gradient descent) for x* which minimizes J(yhat) where J is a known function: this is most common in surrogate-based optimization. The neural-adjoint method is clearly a special case of this latter scenario. See: - "Automatic Chemical Design Using a Data-Driven Continuous Representation of Molecules", - "Multiscale topology optimization using neural network surrogate models", - "Amortized Finite Element Analysis for Fast PDE-Constrained Optimization", - "Conditioning by adaptive sampling for robust design", - the surrogate-based optimization portions of "Algorithms for Optimization". I think the framing of this method as novel and the introduction of a name for the method is inappropriate without significant methodological differences from prior work. I am far from an expert in these areas so may not have picked the best references. I do like the method itself of training a NN surrogate and finding the points which optimize the loss according to the surrogate (which has been demonstrated to be effective in these other problems). The paper should clearly discuss the relation of the proposed method to other work in model-based optimization (making it clear that this is a direct application of NN surrogate modeling to the inverse modeling problem). Possibly it should remove the "neural-adjoint" method branding as this branding is misleading in making the proposed method seem like a newly derived thing. The paper should also relate the techniques in section 3.1 to closely related techniques in safe exploration / safe Bayesian optimization (staying on the manifold of designs).

Reproducibility: Yes

Additional Feedback: --- Update after rebuttal: Thanks for the effort you put into the response. I think if the boundary loss is a main contribution of the paper, there needs to be more insight into its design. Currently it looks quite ad-hoc (just a loss to force the x to be within 2 sigma of the mean) and it would be good to understand the effect of this on different datasets, how it varies with the number of sigma from the mean, hard/soft threshold etc. It's worth noting that there is somewhat similar work on avoiding unrealistic x in surrogate-based optimization, albeit mostly in model-based RL: see model-ensemble TRPO, or model-predictive policy learning with uncertainty regularization. These seem to me to be more principled, in a sense, than using the mean and standard deviation of the training data. I appreciate the work and effort you put into benchmarking on a number of problems and I think this is of great value. I maintain my score, but I hope that if this is not accepted you will resubmit (e.g. to ICLR), with a rewrite to adjust or remove the "neural adjoint" branding and position your work more clearly w.r.t. model-based optimization, and with more insight into the boundary loss, and understanding of this boundary loss vs other techniques in uncertainty-aware / safe / robust optimization. ---


Review 2

Summary and Contributions: This paper proposes a new method, a new performance metric, and presents a benchmark evaluation on four tasks of inverse problems, where one of them is newly constructed in this paper. The contributions of this paper are three folds. - A simple yet strong method to inverse problems. - A new metric that characterizes the performance trade-off in terms of inference efficiency. - A new benchmark dataset constructed from training a network ensemble on 40k examples to fit the simulator of meta-material design.

Strengths: - The proposed inverse solver is simple yet accurate, which serves as a strong baseline method for future research. - The proposed evaluation metric looks reasonable. - The proposed benchmark evaluation looks fair and thorough, which builds a foundation for future research.

Weaknesses: This paper focuses only on measuring the error of point estimates while lacking the error of the estimated “posterior probability distributions,” which has been done in [1,2]. Specifically, [1] used calibration error, and [2] used MMD as the distribution-based metric. In this paper, the “Inverse model performance metrics” subsection in Related Work (line 102-106) only mentioned MSE of point estimates. Throughout the paper, the keyword “posterior” only appears once in Related Work. Since one of the focus of this paper is the evaluation metric, the lack of discussion, comparison, and experiments on distribution-based metrics pose a weakness in this paper. Section 4 can be improved in terms of technical writing. Specifically, I cannot understand the four methods without reading their original papers since the technical description is not precise and clear enough. The main reason that I cannot understand clearly is due to the somewhat simplified equations (Eq. 6, 7, 8, and 9). For example, all the four equations start with “Loss” as their left-hand side, which is too simplified since there should have the model parameters to be learned. Furthermore, the actual expression of \hat{x} or \hat{y} in each method is not revealed. For readers not familiar with the area, such simplification may make the reading harder. In line 94-96: “Earlier work on Mixture density networks [14] model the direct conditional distribution using gaussians. However due to low accuracy reported by [2], we do not include comparison to this work.” I am not sure whether it is an adequate statement. In [2]’s Figure 4 and 5, Mixture density networks (MDN) [14] performs quite competitively. I would like to know why this paper ignores the comparison with [14].

Correctness: Most of the claims and method in the paper looks correct. The empirical methodology looks correct, too.

Clarity: The clarity of Section 4 has room for improvement. The overall clarity is only borderline to me. I cannot have a clear picture after the first-round reading, though I finally understand most of the paper after the third-round reading.

Relation to Prior Work: From my perspective, this paper looks closely related to [1] and [2]: [1] Analyzing inverse problems with invertible neural networks, ICLR19. [2] Benchmarking invertible architectures on inverse problems, ICML19 workshop. The difference between this work and [1,2] can be seen from the contribution summary (line 68-80) and the related work (line 97-101). Therefore, the relation to prior work is clear enough to me.

Reproducibility: Yes

Additional Feedback: The “Tandem model” subsection (start from line 164) is hard to understand. Comparing with the supplementary material, are the role of “encoder” and “decoder” reversed in line 164-166 in the main paper? It would be better to explain in the caption of Figure 3 about why Figure 3(d) has no yellow line (INN). I know that the supplementary material (line 151) has mentioned that. It is better to cite [1] as an ICLR19 paper, not arXiv. Similarly, [2] should be cited as an ICML19 workshop paper. Current citation makes it look like a conference paper. In the supplementary material, “Parameters” in Table 2 should be called “Hyperparameters.” In the supplementary material, I cannot understand how to define the performance when T=0 (subsection 6.1). It would be better to explain clearly how each method is applied with T=0. In the supplementary material, Figure 3 is hard to understand. Why the upper-left of the left subfigure is white? Why the right of the middle subfigure represent x, not \hat{x}? Why the upper-left of the middle subfigure is y, not y_{gt}? Typos: Main paper line 128: “be a our” Main paper line 167-168: “but we found that adding in our boundary loss (see 3.1),” Main paper line 246: “found found” Supplementary material line 95: “from from” ======== post rebuttal ======== Thanks for the rebuttal. Table 1 in the rebuttal well addressed my first concern in my Weaknesses session. My 2nd concern (about writing) is not mentioned in rebuttal, which I can understand it is due to space limit. My 3rd concern is addressed and I can understand your situation. I really appreciate the author response, especially Table 1. However, after reading all the reviews and the rebuttal, I am afraid that I cannot give an overall score of 7 (a good submission, accept) because (1) The writing of the technical methods (as my second concern) is not ready to me. (2) As mentioned by R1, it seems that this work is somewhat similar to some particular works, and a major revision of the introduction and related work seems to be required for clarifying and positioning this work. Overall, the main reason this paper is not ready to me is mostly about writing.


Review 3

Summary and Contributions: This paper proposes a neural inverse method that first learns a forward model and then uses SGD with random initializations to find the inverse. A boundary loss keeps the inverse solutions close to the training data. The authors propose an evaluation based on the number of samples taken and propose two new benchmark tasks.

Strengths: This method can be effective at solving one-to-many inverse problems when the forward model can be well approximated by a neural net. This sidesteps the challenge of learning one-to-many inverse functions. Moreover, the inverse function doesn’t need to keep all its information in the weights, instead the method just requires a good approximation of the forward model and the remaining optimization can be done using SGD.

Weaknesses: The boundary loss imposes a specific prior on the inverse solutions and assumes a specific data distribution — that the dimensions of x are independent and concentrated within 2 standard deviations of the mean. It would be hard to imagine this boundary loss working effectively in more complex data regimes. Moreover, this paper is primarily concerned with evaluating the re-simulation error with no evaluation of the actual inverses, $x$, such as whether the posterior $p(x|y)$ can resemble that of the training distribution.

Correctness: If re-simulation error is the only metric that you care about, then this method is generally correct. Approximating the forward model with a neural net and using SGD with random initializations can be a reasonable solution. The only concern is whether the boundary loss could result in a poor solution, i.e. you can imagine a $y$ which requires some $x$ outside of the $2\sigma$ range but boundary loss outweighs the L2 loss.

Clarity: The paper is generally well written. Some descriptions of the comparison models in section 4 have simplified losses and it’s not clear what the general loss is and what the learned parameters are.

Relation to Prior Work: The paper offers a reasonable discussion of prior work.

Reproducibility: Yes

Additional Feedback: Update after author response: Generally I agree with the points made by reviewer 1 -- in the context of the prior work, the boundary loss and evaluation are somewhat incremental contributions. I would have liked to see a more generalizable boundary loss/prior and more detailed evaluation that prior work has done.


Review 4

Summary and Contributions: The paper has following contributions: 1. A neural adjoint method for inverse problems which outperforms other strong baselines of [4,5], cVAE, INN methods. Briefly, this method entails training a forward approximator "f" on data (x,y), and then use of gradients for different initializations (x_0) to obtain the input x for given y. 2. It proposes a benchmark with many old/new inverse problems and compares the proposed adjoint method to the baselines. They perform comparisons of inverse model with multiple samples (or over inference time). with the "r_t" metric.

Strengths: + Experimental evaluation is a strength of this work. The proposed method is compared to many relevant baselines in the literature. All the methods are evaluated on different inverse problems and their runtime and accuracy are measured. + The proposed neural adjoint method with enough inference budget obtains the best accuracy on the benchmark.

Weaknesses: - Unlike Tandem Model [4,5] and cVAE based methods the proposed method uses gradient updates and therefore is slow. The authors acknowledge this in the manuscript and demonstrate study the method as a function of inference budget. - The sampling performed to obtain different initializations x_0 seems important for the convergence to optimum. This is not experimentally evaluated carefully on the proposed benchmarks, except for Tab. 1 in supplementary where it is compared to sampling from uniform distribution.

Correctness: The contributions of the method in the form of a benchmark for inverse problems and the experimental evaluation of proposed NA method and other baselines cVAE, INN, [4,5] is technically correct and sound.

Clarity: The paper is well written and easy to follow. Some important ablation results (Tab. 1) are in supplementary and can be moved to the main paper.

Relation to Prior Work: Prior work on inverse problems is discussed and included in the benchmark of the paper.

Reproducibility: Yes

Additional Feedback:

[Author Response · NeurIPS 2020]

**Author Response for Paper 6449** We thank the reviewers for their time and helpful feedback! Below we summarize and respond to as many major reviewer comments as space permits. Please note that any in-text references use the same numbering as the manuscript.

**1.The neural-adjoint appears similar to some recently-proposed methods. The authors should clarify the degree of its novelty, and reconsider its branding:** Reviewer #1 shared several recent publications and, after reviewing these publicataions (and related works that we found), we agree that several of these papers do propose methods that are fundamentally similar to the Neural Adjoint (NA). We thank the reviewer for bringing these relevant publications to our attention.

Given this new literature however, we find that our original claims of novelty are still valid. In our manuscript (Section II) we explained that the Neural-Adjoint (NA) is based directly upon the approach proposed in [21]. The approach in [21] and the NA share the following methodology: a deep neural network is used as a surrogate for the forward model, and gradient descent is used to optimize the forward model output with respect to its input. In Section II we claimed two main novelties with respect to [21]: (i) we propose the boundary loss and $\Gamma$ design that result in the NA method, and (ii) we provide thorough empirical evidence that the NA is competitive or superior across many inverse tasks. We find that the new references, and their proposed methodologies, share (*roughly*) the same similarities and differences with respect to the NA as [21]. And therefore our original claimed contributions are still valid. As reviewer 1 noted though, the NA does represent a specific variation on a broader class of recently-proposed methods - more than just reference [21].

Given the expanded related work, however, we do agree we should revise our NA branding. We still believe it is reasonable to employ a name, in this case "Neural Adjoint", so that our method can be easily referenced both within and outside of our paper. However, we will revise our manuscript to expand the related work and clarify that the NA method has major methodological similarities with many recently proposed methods, explain the similarities/differences, and explain that NA is the name for our particular variation on this general class of recent methods.

**2.The authors should report the posterior distribution of solutions (e.g., Maximum Mean Discrepency measurement (MMD):** We did not include MMD largely to control the scope/length of our paper. However, we agree with the reviewer's that MMD results would be valuable, and we provide them in Table 1. Note that D4: Meta-material is not included due to the intractability of rejection sampling. We find that cINN always has the best (lowest) MMD, closely followed by NA. Due to space limitations, we will need to add Table 1 and accompanying explanations to the appendix. We will briefly summarize these results in the manuscript.

**3.Assumptions and constraints imposed by the boundary loss, $\mathcal{L}_{bnd}$.** We found that our trained neural networks produced poor forward predictions outside of the training data domain. Therefore $\mathcal{L}_{bnd}$ was designed to encourage the model to seek solutions within the training data domain (i.e., where the forward model is accurate), at the cost of limiting the solution space. As shown in the appendix, this tradeoff consistently yielded substantial performance improvements. In all of our benchmark problems, the training data sampling space is

Table 1: Posterior matching MMD score

| Data | NA | TD | cVAE | INN | cINN |
|------|------|------|------|------|------|
| D1 | 0.07 | 2.62 | 0.07 | 2.03 | **0.04** |
| D2 | 0.04 | 2.84 | 0.03 | 1.07 | **0.03** |
| D3 | 0.06 | 2.70 | 1.62 | 0.11 | **0.04** |

well approximated by a hyper-cube of the form $|\hat{x} - \mu_x| - 2\sigma_x$, motivating our $\mathcal{L}_{bnd}$ design. However, we agree with reviewers that this imposes a specific prior on the solutions e.g., a high uniform prior within the hypercube. We also agree that this approximation will *not* work for more complex distributions (e.g., non-convex). We do believe $\mathcal{L}_{bnd}$ can be extended to such cases e.g., via kernel density estimates, and other methods. Based upon reviewer feedback, we will revise our manuscript to clarify these assumptions of our methodology for readers so they understand its limitations.

**4.Contribution beyond prior meta-material work in [24]** In [24] the authors considered one inverse approach, and did not quantify its performance. Here we quantitatively compare five methods. In [24] the single inverse model predictions were simulated with the real electromagnetic simulator, which is very slow. This inspired us to propose to train a high accuracy neural simulator (we achieve MSE of 6e-5 here versus 1e-3 in [24]) and treat it as the real simulator (e.g., sampling training data from it, and evaluating re-simulation errors using it), which was not done in [24]. This allows us to easily share our neural simulator with other groups (also not done in [24]) and for others to easily replicate and build upon our study of this modern problem.

**5.Mixture density network (MDN) not included for comparison** We did not include the MDN because we think performance comparisons with the cINN and INN are sufficient to support our conclusions. However, we do agree with the reviewer that inclusion of MDN would strengthen our conclusions. However, we did not have time to code, quality assure, and test the MDN on all datasets within the rebuttal period. Therefore we will be unable to address this limitation.

[Meta-Review · NeurIPS 2020]

The reviewers all liked some aspects of the paper, agreed that the presentation of the technical part can be improved (e.g just spotted a missing ^2 in eq 7) and that the main contributions are 1) a heuristic (eq 5) to regularise the inverse solutions to be close to the input data and 2) the empirical evaluation suite. The presentation can (and should) of course be improved but it is not likely that the method will change if the paper is rejected now and resubmitted to another conference. Therefore acceptance is recommended together with a strong encouragement to rework the presentation for the final version.